# Economic evaluation of diagnostic tests for Thai patients with tuberculosis: A dynamic transmission model approach

Natthakan Chitpim[1], Naiyana Praditsitthikorn[2], Lisa J. White[3], Aronrag Meeyai[4,5], Jiraphun Jittikoon[6], Kornkanok Bunwong[7,8], Sitaporn Youngkong[9,10], Montarat Thavorncharoensap[9,10], Surakameth Mahasirimongkol[11], Wanvisa Udomsinprasert[6], Usa Chaikledkaew[9,10¤]*

1 Social, Economic and Administrative Pharmacy (SEAP) Graduate Program, Faculty of Pharmacy, Mahidol University, Bangkok, Thailand, 2 Department of Disease Control, Ministry of Public Health, Nonthaburi, Thailand, 3 Department of Biology, University of Oxford, Oxford, United Kingdom, 4 Centre for Tropical Medicine and Global Health, Nuffield Department of Clinical Medicine, University of Oxford, Oxford, United Kingdom, 5 Department of Epidemiology, Faculty of Public Health, Mahidol University, Bangkok, Thailand, 6 Department of Biochemistry, Faculty of Pharmacy, Mahidol University, Bangkok, Thailand, 7 Department of Mathematics, Faculty of Science, Mahidol University, Bangkok, Thailand, 8 Centre of Excellence in Mathematics, MHESI, Bangkok, Thailand, 9 Social Administrative Pharmacy Division, Department of Pharmacy, Faculty of Pharmacy, Mahidol University, Bangkok, Thailand, 10 Mahidol University Health Technology Assessment (MUHTA) Graduate Program, Mahidol University, Bangkok, Thailand, 11 Information and Communication Technology Center, Office of the Permanent Secretary, Ministry of Public Health, Nonthaburi, Thailand

¤Current Address: Social and Administrative Pharmacy Division, Department of Pharmacy, Faculty of Pharmacy, Mahidol University, 447 Sri-Ayudhaya Road, Rajathevi, Bangkok 10400, Thailand
* usa.chi@mahidol.ac.th

## Abstract

Conventional tuberculosis (TB) diagnosis is time-consuming, while newer molecular assays such as Xpert MTB/RIF and loop-mediated amplification test for TB (TB-LAMP) provide faster results but at a higher cost compared to sputum smear microscopy (SSM) with culture and drug susceptibility testing (DST) in Thailand. This study assessed the cost-utility of TB diagnostic algorithms as either initial or add-on tests from a societal perspective for TB diagnosis in the general Thai population. A dynamic transmission model was employed to evaluate five TB diagnostic algorithms over a 15-year period. Costs were calculated in 2023 Thai Baht, with results presented as incremental cost-effectiveness ratios (ICERs) compared to SSM with culture and DST. One-way and probability sensitivity analyses were conducted to assess parameter uncertainty. Compared to SSM with culture and DST, the ICER values (Baht per QALY gained) of TB-LAMP Add-On (3,563), Xpert MTB/RIF Add-On (3,670), and TB-LAMP Initial (6,429) indicated that these algorithms were cost-effective, while Xpert MTB/RIF Initial emerged as a cost-saving option. One-way sensitivity analysis results revealed that the utility of the first-line treatment exhibited the highest variability in ICERs, followed by the unit cost of Xpert MTB/RIF. The results supported the adoption of Xpert MTB/RIF as an initial test for the general Thai population. These findings provide evidence for policymakers to integrate molecular testing into Thailand's Universal Coverage Scheme benefit package, aligning with national TB strategies to reduce TB incidence and mortality.

**Data availability statement:** All relevant data are within the manuscript and its Supporting Information files.

**Funding:** We would like to acknowledge the Thailand Science Research and Innovation under the Ministry of Higher Education, Science, Research and Innovation for providing the Royal Golden Jubilee Ph.D. scholarship for NC and UC (grant no. PHD/0104/2559). This research project has been funded by the Health System Research Institute (HSRI), Thailand as well as Mahidol University (Fundamental Fund: fiscal year 2024) by National Science Research and Innovation Fund (NSRF). The funders had no role in study design, data collection and analysis, decision to publish, or preparation of the manuscript.

**Competing interests:** The authors have declared that no competing interests exist.

## Introduction

Tuberculosis (TB), primarily caused by *Mycobacterium tuberculosis* (MTB), predominantly affects the lungs (80%) [1,2]. In 2022, the World Health Organization (WHO) reported 10.6 million TB diagnoses, with 1.1 million fatalities worldwide [3]. Thailand, an upper-middle-income country [4] with a high TB and TB-HIV burden [3], reported an estimated TB incidence rate of 155 cases per 100,000 population and a TB mortality rate of 18.9 deaths per 100,000 population in 2022, with a 2.33% prevalence of multidrug-resistant TB (MDR-TB) [5]. Only 70% of TB cases were reported, with 83% receiving treatment in 2020 [5], primarily due to underdiagnosed TB patients [3]. Early TB diagnosis improves patient survival, quality of life, and reduces economic burdens [6].

The Sustainable Development Goals and the WHO's End TB Strategy serve as global benchmarks for TB control and elimination [7]. To meet the 2030 targets, Thailand must achieve an annual TB incidence reduction of 12.5%, decreasing from 171 cases in 2014 to 88 cases per 100,000 population in 2030. Recent data indicate a 2.7% annual reduction over the past decade [1,5]. The Thai national plan prioritizes early TB diagnosis through the use of molecular diagnostics for all presumptive TB cases [1].

Conventional TB screening in Thailand mainly utilizes Chest X-Rays (CXR) and sputum smear microscopy (SSM). SSM, despite its limited sensitivity which can lead to misdiagnosis (46%), is valuable in high prevalence areas [8]. Culture and drug susceptibility testing (DST) are employed to confirm TB and MDR-TB in cases with negative smears [1]. While conventional methods are cost-effective, they require over four weeks for culture results and six weeks for DST results, contributing to treatment delays and higher mortality (hazard ratio 1.59, 95% CI 1.01 to 2.48) [9]. Currently, there is an urgent need for the WHO-endorsed molecular tests, such as Xpert MTB/RIF, loop-mediated amplification test for TB (TB-LAMP), and line probe assay (LPA), which offers automated nucleic-acid amplification tests (NAATs) [3]. These rapid diagnostic methods require fewer resources, occupy less space, and yield results within 1–2 days [10], thereby enhancing TB case identification, facilitating timely treatment, and reducing TB morbidity and mortality [2].

However, molecular testing is costly and currently only reimbursed under the Civil Servant Medical Benefit Scheme (CSMBS) for government officers and their dependents. It is not covered by the Social Security Scheme (SSS) for private employees or the Universal Coverage Scheme (UCS) for general Thai population. Additionally, previous economic evaluation studies have yielded mixed results when comparing molecular testing to conventional methods, i.e., SSM with culture and DST [11,12]. One Thai economic evaluation study, which utilized a decision tree with a Markov model, indicated that TB-LAMP as the initial test was cost-saving, while Xpert MTB/RIF as the initial test, as well as TB-LAMP or Xpert MTB/RIF used as add-on tests were cost-effective compared with conventional methods [13]. However, these static models did not account for the interactions between individuals and the evolving dynamics of TB transmission over time, which can be critical factors for TB. Policymakers also require robust economic evaluation evidences to make informed decisions regarding which molecular testing methods should be included in health insurance benefit packages. To accurately capture the impact of diagnostic tests on the overall epidemiology of TB within a population, including reductions in transmission rates and changes in the incidence and prevalence of the disease, dynamic transmission models should be employed. Therefore, this study aimed to evaluate the cost-effectiveness of molecular testing in the Thai context using a dynamic transmission model.

## Materials and methods

### Study design

Cost-utility analysis using a dynamic transmission model was employed to compare the costs and quality-adjusted life years (QALYs) of the Xpert MTB/RIF and TB-LAMP algorithms

with current TB diagnosis practices within the general population aged 15 years and older. These suspected patients were not categorized as either high-risk or drug-resistant populations, as outlined in the 2018 Thailand National Tuberculosis Control Program (NTP) guidelines [1]. The calibrated mathematical model simulated disease progression over a 15-year time horizon, utilizing a 1-month cycle length to align with the natural history of TB. This model adopted from Menzie et al. [2] was specifically tailored for evaluating TB diagnostics within the Thai context incorporating various scenarios relevant to the country. The analysis was performed using Program R version 4.3.1 (2023-06-16).

## Ethics approval

Our study used data related to quality of life as well as direct medical and non-medical costs from published studies. The Institutional Review Boards (IRB) of Mahidol University approved this study (COA.No.MU-DT/PY-IRB 2019/002.0701) through the expedited review procedure. The need for informed consent for this study was waived by the ethics committees, as all data were obtained from published studies. All procedures performed in the study were in compliance with international guidelines for human research protection such as Declaration of Helsinki, the Belmont Report, CIOMS Guidelines and the International Conference on Harmonization in Good Clinical Practice (ICH-GCP).

## Diagnosis strategies

This study evaluated five TB diagnostic algorithms designed based on the 2018 guidelines of the NTP of Thailand [1].

**Algorithm 1: SSM with culture and DST.** In Algorithm 1, individuals started in a susceptible state and could progress to latent tuberculosis infection (LTBI) or active TB infection. The diagnostic process began with SSM, classifying individuals as either positive (POS) or negative (NEG) test for TB. Those who tested positive via SSM would undergo DST to determine the appropriate treatment regimen. If the treatment was successful, the individual returned to an LTBI state. However, if treatment failed, the patient was re-diagnosed to determine the next steps. Depending on the results, they might be classified as treated multidrug-resistant TB (MDR-TB) or extensively drug-resistant TB (XDR-TB). Patients diagnosed with MDR-TB followed a specific treatment regimen designed for this condition. Upon successful completion of treatment, these patients returned to an LTBI state. Similarly, patients diagnosed with XDR-TB followed an appropriate regimen and were also assumed to return to an LTBI state after successful treatment, with the assumption that there were no further failures.

**Algorithm 2: Xpert MTB/RIF add-on.** Similar to Algorithm 1, individuals in this model progressed through the state of Susceptible, LTBI, POS, and NEG states. In the POS state, patients were first diagnosed by SSM, and then treated based on DST results. In the NEG state, patients were reconfirmed with Xpert MTB/RIF. Depending on the results, they were directed to appropriate treatment pathways. Those with positive TB results but negative rifampicin (RIF) resistance were moved to "Treated TB" state, while those with both TB and RIF positivity were directed to the "Treated MDR." The progression mirrored the process described in Algorithm 1.

**Algorithm 3: TB-LAMP add-on.** Individuals transitioned from Susceptible to LTBI, POS, or NEG. POS patients would follow Algorithm 1. In the NEG state, patients who tested positive with TB-LAMP underwent DST to determine the appropriate treatment selection.

**Algorithm 4: Xpert MTB/RIF initial.** Xpert MTB/RIF replaced SSM as the initial diagnostic test for TB. Positive results from Xpert MTB/RIF led to direct treatment without requiring DST confirmation. Negative results placed individuals in the NEG state, where they could potentially be re-infected and undergo the diagnostic process again.

**Algorithm 5: TB-LAMP initial.** In this algorithm, individuals progressed through susceptible, LTBI, POS, and NEG states. POS individuals were diagnosed using TB-LAMP and then underwent DST. Treatment was based on DST results, leading to classification and management as either treated TB, treated MDR-TB, or treated XDR-TB. Those in NEG state were not treated immediately but could return for diagnosis if symptoms developed. Patients in treated TB or treated MDR-TB states followed a progression similar to that described in Algorithm 1.

## Model structure

The model, illustrated in Fig 1, outlines the progression of TB disease among suspected patients. Initially, individuals entered a susceptible state, from which they might transition to LTBI, POS, or NEG. The initial diagnostic approach could be one of three tests: SSM, Xpert MTB/RIF, or TB-LAMP. Once in the POS state, patients were subject to treatment based on DST. Those with pan-sensitive TB received a 6-month regimen, i.e., 4 months of isoniazid, rifampicin, pyrazinamide, and ethambutol, followed by 2 months of isoniazid and rifampicin (4IRZE-2IR), after which successful treatment resulted in a return to LTBI. Patients diagnosed with MDR-TB underwent a 9-month regimen, i.e., 4 months of amikacin, levofloxacin, ethionamide, clofazimine, isoniazid, and ethambutol, followed by 5 months of levofloxacin, clofazimine, ethambutol, and isoniazid without amikacin (4 Am-Lfx-Eto-Cfz-I-Z-E/5 Lft-Cfz-E-Z). Successful treatment led to LTBI, whereas incomplete treatment might result in progression to XDR-TB. XDR-TB patients followed a 2-year regimen consisting of 6 months of capreomycin, linezolid, moxifloxacin, clofazimine, and bedaquiline, or ≥ 2 months of capreomycin, linezolid, moxifloxacin, and clofazimine, or ≥ 12 months of linezolid, moxifloxacin, and clofazimine (6 Cm Lzd Mfx Cfz Bdq/≥ 2 Cm Lzd Mfx Cfz/≥ 12 Lzd Mfx Cfz), and upon successful completion, they transition back to LTBI, assuming no treatment failures.

In contrast, individuals in the NEG state initially did not have TB but might require additional diagnostic tests, such as SSM with culture, Xpert MTB/RIF, or TB-LAMP, for further confirmation. If reclassified as POS, they were treated according to the same protocols as initially diagnosed patients. The model also accounted for spontaneous recovery, where some individuals might revert from LTBI to a state of no active disease. Additionally, patients experiencing treatment failures were re-diagnosed and potentially reassigned to different treatment pathways based on updated diagnostic results.

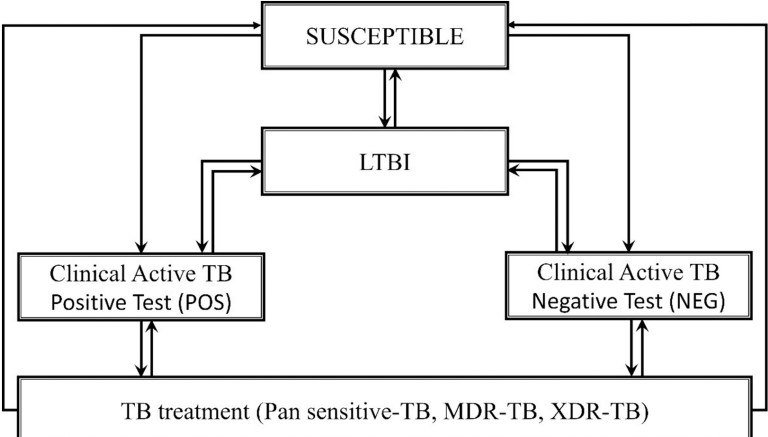

**Fig 1. A dynamic transmission model.** TB; tuberculosis: MDR-TB; multidrug-resistant tuberculosis: XDR-TB; extensively drug resistant tuberculosis.

## Model parameters

The model incorporated input parameters such as probability of various variables, including the sensitivity and specificity of TB diagnostic methods, transition probabilities between health states, costs and health utilities (detailed in S1 Table). The analysis in this study was informed by data from systematic reviews of randomized controlled trials (RCTs) and meta-analyses.

## Transitional probabilities

The calculation of transitional probabilities for treatment outcomes, including cure, mortality, and loss to follow-up, was based on data from the WHO's Global TB Report 2021 [3]. Furthermore, information on TB-related parameters, such as the percentage of suspected TB patients showing abnormal CXR or clinical symptoms, as well as the prevalence of drug-sensitive TB (DS-TB), MDR-TB, or XDR-TB within each intervention, was obtained from the National Tuberculosis Information Program (NTIP), Division of Tuberculosis, Ministry of Public Health, Thailand [4]. The sensitivity and specificity of diagnostic tests were extracted from both WHO guidelines and meta-analysis studies [6].

## Costs

This economic evaluation was conducted from a societal perspective, which included both direct medical costs (DMCs) and direct non-medical costs (DNMCs). All costs were presented in 2023 Baht values, adjusted using the Thailand Consumer Price Index (CPI) for medical care [14]. DMCs covered expenses related to TB disease and drug treatment, including screening and diagnostic costs, e.g., Xpert MTB/RIF, TB-LAMP, CXR, SSM, culture, and DST. These costs were sourced from the Thai standard cost list for health technology assessment (HTA) [15] and the Comptroller General's Department of Thailand [16], with adjustments using a cost-to-charge ratio of 1.63 [17]. Treatment costs encompassed medication expenses, aligned with the Thai NTP guidelines (e.g., 2HRZE/4HR, 4Km Mfx Pto Cfz Z E H/5 Mfx Cfz Z E, 8 Cm 12Lzd 20Cfz 20Mfx 6Bdq), as well as outpatient, inpatient, and healthcare service costs, also retrieved from the Thai standard cost list for HTA [15]. DNMCs, which included expenses for transportation, food, accommodation, and formal care, were derived from the study by Youngkong S, et al. [16,18].

## Health utilities

We utilized health-related quality of life data from TB patients in Thailand to calculate within the dynamic model. In Thailand, a study conducted by Kittikraisak W. et al. [19] collected utility data using the EQ-5D-5L instrument among TB patients in 2012. It was assumed that these utility data were obtained after the patients had been diagnosed with active TB, including both pan-sensitive and drug-resistant TB, and were currently undergoing treatment. These utilities were categorized into those for patients undergoing treatment for pan-sensitive TB, MDR-TB, XDR-TB, and those who had been cured of TB. Based on the findings of Kittikraisak W. et al. [19], it was further assumed that the utility for XDR-TB was equivalent to that of MDR-TB.

## Uncertainty analysis

To account for uncertainties related to treatment costs and influential parameters in the cost-effectiveness analysis, we conducted one-way sensitivity analysis and probabilistic sensitivity analysis (PSA). In the one-way sensitivity analysis, we altered specific variable values to assess their impact on cost-effectiveness results, specifically focusing on the incremental cost-effectiveness ratios (ICERs), calculated by dividing the difference in costs by the difference in QALYs. Transitional probabilities were adjusted by 95% confidence intervals (CI) or

10% when CI data were unavailable. Cost parameters were varied by 25%, while the discounting rate ranged from 0% to 6%. The results of one-way sensitivity analysis were visually represented using a tornado diagram, highlighting the parameters that most significantly influenced the cost-effectiveness outcomes.

The PSA was conducted using Monte Carlo simulation with 1,000 iterations. The method introduced random variations in parameter values based on their respective distributions to assess the impact of uncertainty on the economic evaluation results. The results of the PSA were depicted through cost-effectiveness planes and cost-effectiveness acceptability curves, providing a visual representation of the probability that an intervention was cost-effective at different willingness-to-pay (WTP) thresholds. According to the Thai HTA guidelines, a societal WTP threshold in Thailand was set at 160,000 Baht per a QALY gained [17]. This threshold is used to determine whether an intervention is considered cost-effective in the Thai context.

### Model validation

Model calibration employed Bayesian estimation to align model parameters with existing WHO data on TB incidence [3] and mortality in Thailand [4], as well as newly generated model data. The calibration process involved iteratively adjusting model parameters to ensure that the model accurately represented the observed data. This was achieved by generating samples from the posterior distribution of model parameters through repeatedly sampling from a proposal distribution, with each sample evaluated based on specific criteria. For the calibration, prior distributions for the parameters were defined, along with a likelihood function to assess how well the model fit the data. Posterior parameter samples were then generated, allowing for the computation of summary statistics, such as means and 95% credible intervals. These statistics were crucial for evaluating the model's performance and making predictions about future observations.

## Results

### Clinical outcomes

Fig 2 illustrates the trend in TB disease incidence cases over time, as diagnosed using various diagnostic algorithms. The analysis revealed that the Xpert Add-On algorithm resulted in the highest number of diagnosed cases, totaling 58,293 cases equivalent to 102 cases per 100,000 population). This was followed by the TB-LAMP Add-On algorithm, identifying 57,372 cases or 101 cases per 100,000 population. The Xpert Initial algorithm accounted for 47,364 cases, translating to 83 cases per 100,000 population, while the TB-LAMP Initial algorithm reported 50,338 cases or 88 cases per 100,000 population.

When compared to these newer algorithms, the conventional method showed the highest incidence rate, with 58,995 cases equivalent to 103 cases per 100,000 population. Among the advanced diagnostic methods, Xpert Add-On demonstrated the lowest incidence rate, showing a reduction of approximately 1.44% relative to the conventional method. The TB-LAMP Add-On method followed closely, with a 2.78% reduction. In contrast, the Xpert Initial method exhibited a significant 19.74% decrease, and the TB-LAMP Initial method demonstrated a 14.67% reduction, both when compared to the incidence rate associated with the conventional method. These results highlight the potential of the newer diagnostic algorithms, particularly Xpert Initial and TB-LAMP Initial, in effectively reducing the incidence of TB cases over time when compared to the conventional method.

### Cost-effectiveness analysis

Table 1 presents the cost-effectiveness results. From a societal perspective, encompassing approximately 57 million individuals [20], the cost of diagnosing TB with various methods

in Thailand was assessed. Among the diagnostic methods, Xpert MTB/RIF Initial yielded the lowest total cost of 1,292,192,150 Baht, which represented 20% of the total diagnostic costs. In contrast, TB-LAMP Initial incurred the highest total cost at 1,411,099,374 Baht, constituting 24% of the costs. Xpert MTB/RIF Add-On had a total cost of 1,408,692,323 Baht, equivalent to 14% of the diagnostic costs, while TB-LAMP Add-On totaled 1,352,902,142 Baht, representing 16% of the costs. SSM with culture/DST had a total cost of 1,320,355,676 Baht, accounting for 14% of the diagnostic expenses.

The ICERs of Xpert MTB/RIF Add-On, TB-LAMP Initial, and TB-LAMP Add-On compared to SSM with culture/DST were 3,670 Baht, 6,429 Baht, and 3,563 Baht per QALY gained, respectively. In contrast, Xpert MTB/RIF Initial demonstrated dominace or cost-savings. The Efficiency Frontier was utilized to evaluate the cost-effectiveness of the current diagnostic method (SSM with culture) against alternative molecular diagnostic techniques (Xpert MTB/RIF Add-On, TB-LAMP Add-On, Xpert MTB/RIF Initial, and TB-LAMP Initial) in the general population. Fig 3 demonstrates that the Xpert MTB/RIF Initial algorithm exhibited the most economically favorable option due to cost savings, followed by Xpert MTB/RIF Add-on, when compared to the Thai societal WTP threshold of 160,000 Baht per QALY gained.

## Uncertainty analysis

In this study, one-way sensitivity analysis was conducted to evaluate how individual variables affected the ICER for diagnosing TB in the general population using the Xpert MTB/RIF Initial method. The analysis revealed that the utility associated with the first-line treatment

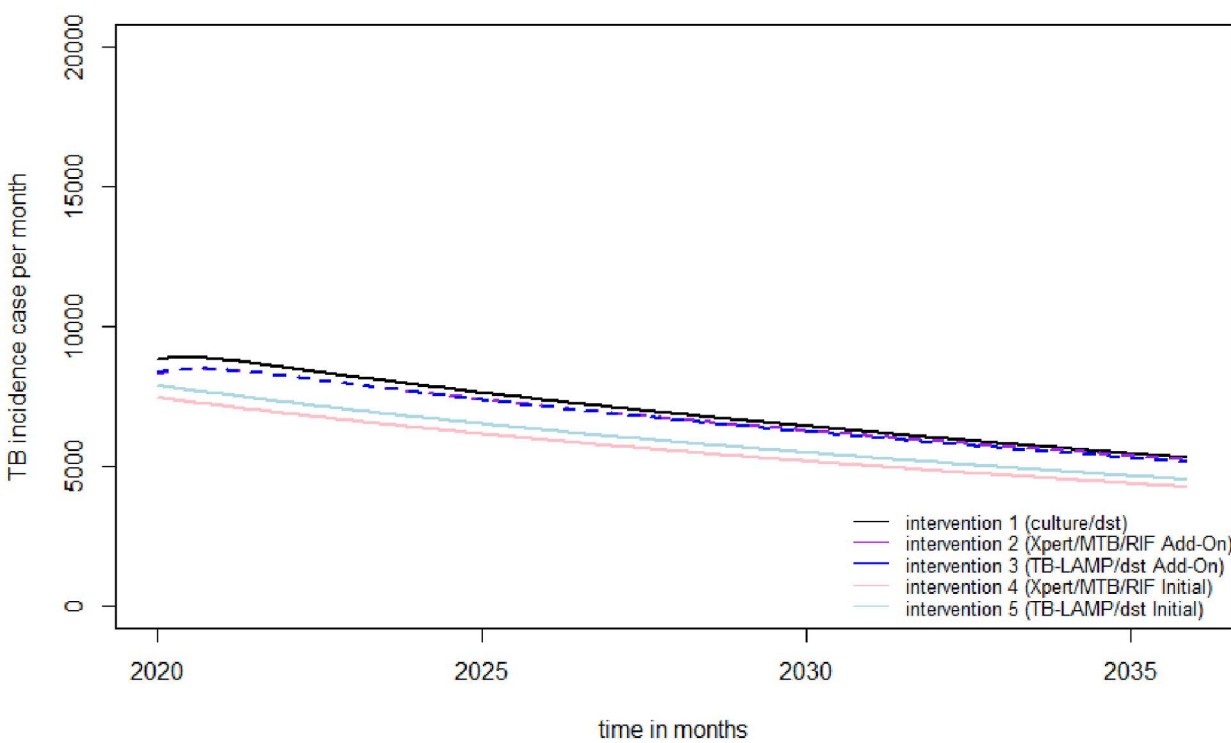

**Fig 2. A predicted incidence monthly TB case by model of each diagnostic TB intervention.**

**Table 1. Cost-effectiveness results.**

| Intervention | Diagnosis cost | Treatment cost | Total cost | Total QALY | Incremental Cost | Incremental QALYs | ICER |
|---|---|---|---|---|---|---|---|
| SSM with culture/DST | 190,102,532 (14%) | 1,134,070,497 | 1,320,355,676 | 63,335,243 | | | |
| Xpert MTB/RIF Add-On | 193,144,979 (14%) | 1,219,813,491 | 1,408,692,323 | 63,359,317 | 88,336,647 | 24,073 | 3,670 |
| TB-LAMP Add-On | 215,355,178 (16%) | 1,141,416,793 | 1,352,902,142 | 63,344,378 | 32,546,465 | 9,135 | 3,563 |
| Xpert MTB/RIF Initial | 256,874,864 (20%) | 1,038,792,345 | 1,292,192,150 | 63,344,914 | -28,163,526 | 9,671 | Dominance[a] |
| TB-LAMP Initial | 342,542,850 (24%) | 1,072,182,052 | 1,411,099,374 | 63,349,358 | 90,743,698 | 14,115 | 6,429 |

[a]Dominance in this context refers to both lower costs and improved effectiveness or increased QALYs.

had the greatest impact on the ICER, with a percent change of 254.44% (Fig 4). Additionally, reducing the cost per unit of Xpert MTB/RIF from 880 Baht to 400 Baht resulted in a substantial decrease in ICER values, up to 91%.

The PSA using Monte Carlo simulation with 1,000 iterations demonstrated that the Xpert MTB/RIF Initial diagnostic approach had a high probability of being cost-effective. Specifically, with a WTP threshold of 160,000 Baht per QALY gained, the Xpert MTB/RIF Initial intervention had a 100% probability of being deemed cost-effective (Fig 5).

## Discussion

TB remains a critical global public health issue, with Thailand recognized by the WHO as one of the countries with a high TB burden in 2021 [5]. Despite an estimated 103,000 new TB cases in Thailand, only 70.7% were reported in the TB program's data for the fiscal year 2015 [4]. Traditional diagnostic methods, such as symptom screening, CXR, and SSM, are often delayed, which contributes to the continued spread of the disease [5]. To overcome these

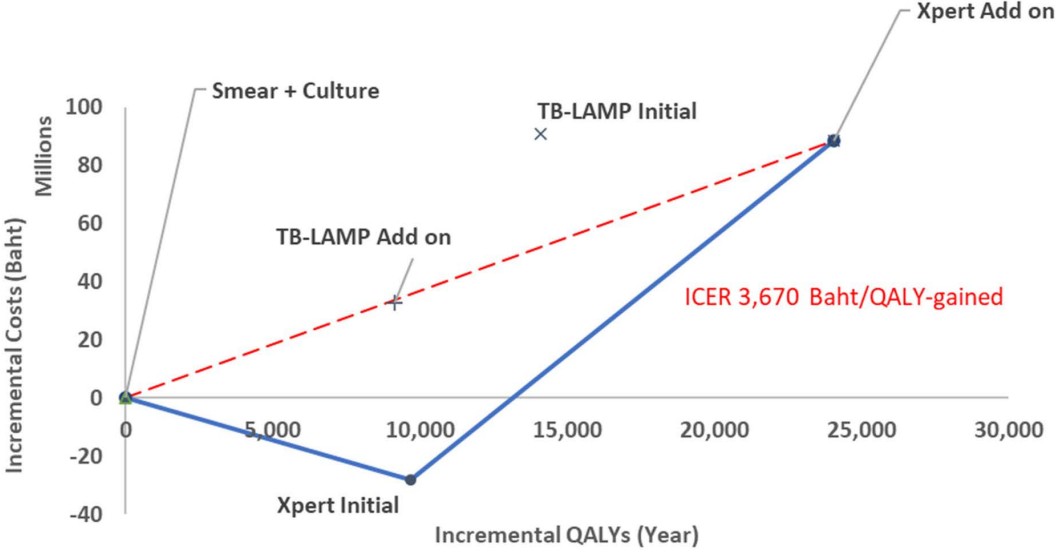

**Fig 3. Efficiency frontier of molecular testing alternatives compared to conventional testing in general population.**

limitations, the WHO has recommended newer diagnostic techniques, such as Xpert MTB/RIF and TB-LAMP, which provide faster TB detection and support global and national TB reduction efforts [5]. However, the high cost of molecular testing presents a challenge for its inclusion in Thailand's national health benefit packages, highlighting the need for comprehensive economic evaluations to guide policy decisions.

At the Thai societal WTP threshold of 160,000 Baht per QALY gained, molecular testing algorithms such as Xpert MTB/RIF Add-on, Xpert MTB/RIF Initial, TB-LAMP Add-on, and TB-LAMP Initial were found to be more cost-effective compared to the conventional method, i.e., SSM with culture and DST in general populations. Univariate analysis identified several key parameters affecting cost-effectiveness, including the utility of pan-sensitive TB during treatment, the unit cost of Xpert MTB/RIF, and the discounting rate of costs. Notably, reducing the unit cost of Xpert MTB/RIF from 800 to 400 Baht led to a significant decrease in the budget impact for diagnosing initial drug-resistant TB across all target groups.

Our findings support the conclusion that molecular testing algorithms are more cost-effective than conventional testing methods. However, our study specifically identified Xpert MTB/RIF as being more cost-effective than TB-LAMP Initial in the general population, which contrasts with the results of our previous economic evaluation that applied a decision tree with a Markov model and found TB-LAMP Initial to be cost-saving [13]. The discrepancy can be attributed to the differences in the modeling approaches used in these two studies. In our current study, we employed a dynamic mathematical model based on Menzies et al. [2] to predict TB transmission and distribution at the population level, taking into account transmission dynamics and the timing of health states. The Xpert MTB/RIF algorithm's ability to simultaneously detect *Mycobacterium tuberculosis* and rifampicin-resistant TB significantly reduces the waiting time for treatment initiation compared to conventional methods and TB-LAMP. In the TB-LAMP algorithm, DST typically requires an additional 2–4 weeks, which can delay the start of treatment. This delay increases the risk of patients being lost to follow-up and results in higher mortality rates [9,21,22].

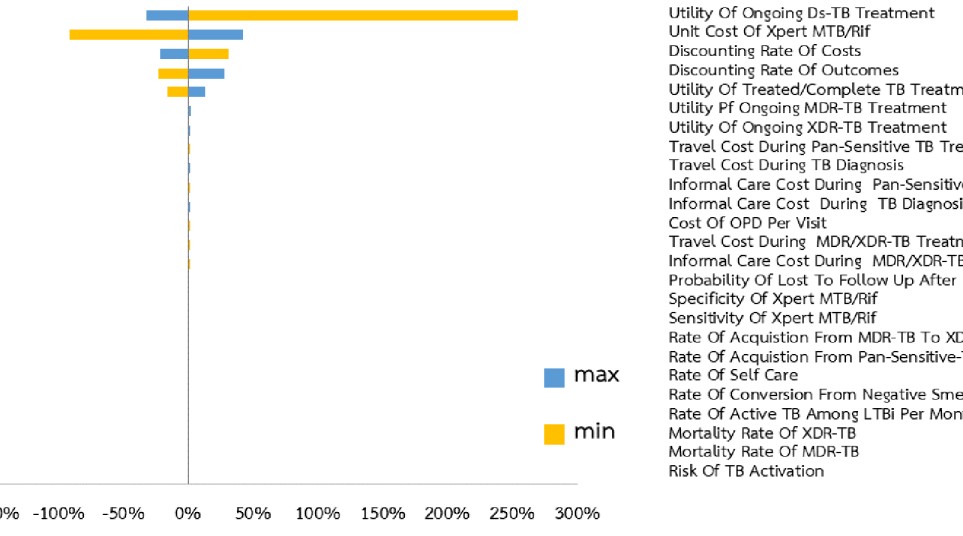

**Fig 4. Tornado diagram.**

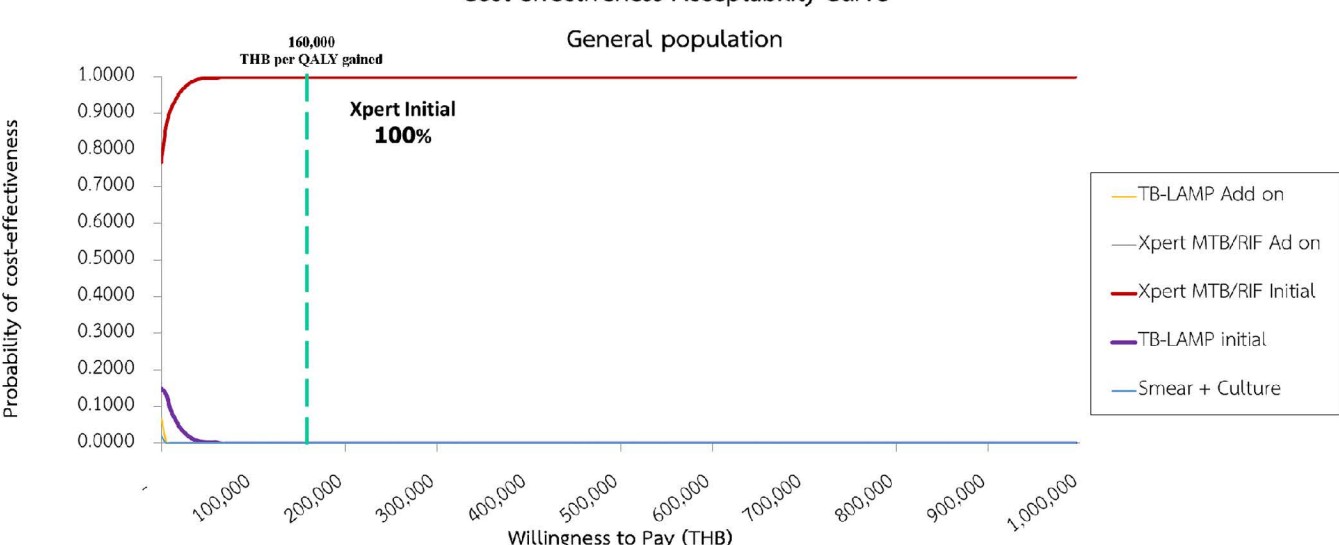

**Fig 5.  Cost-effectiveness plane based on a societal perspective.**

Our findings are consistent with previous economic evaluations of Xpert MTB/RIF, which have consistently demonstrated its superior cost-effectiveness compared to conventional methods such as SSM with culture and DST. For example, a study conducted in Thailand by Khumsri et al. [12] similarly concluded that Xpert MTB/RIF is more cost-effective than SSM. Additionally, research by You JH et al. [23] Wikman-Jorgensen PE, et al. [24] and Menzie et al. [2] further emphasized the cost-effectiveness of incorporating Xpert MTB/RIF to smear-based diagnostic strategies, particularly for cased where sputum smears are negative. These studies reinforce the value of Xpert MTB/RIF as a more efficient alternative to conventional TB diagnostic methods, especially in improving health outcomes and optimizing resource allocation in TB control efforts.

This study differs from previous work in several key aspects. Firstly, we used comprehensive data from the WHO and the NTIP, which allowed for more accurate and realistic representation of TB patients in Thailand [25,26]. Secondly, we incorporated all relevant molecular testing algorithms, including both Xpert MTB/RIF and TB-LAMP, both of which are currently available and applicable within the Thai healthcare context. These algorithms were endorsed by TB clinical experts and stakeholders involved in TB management in Thailand, ensuring that the study's findings are grounded in practical and expert-informed perspectives. Lastly, our study employed a dynamic transmission model, which effectively captured the realistic progression and transmission dynamics of TB disease, offering a more comprehensive understanding of how these diagnostic strategies impact TB control efforts at the population level. This approach contrasts with our previous study which has used static models, thereby providing a more robust and policy-relevant analysis.

However, the study has several limitations. Firstly, it did not include the costs associated with TB re-diagnosis across all health states, potentially underestimating the total economic burden, especially in cases of treatment failure or relapse. Future studies should incorporate these costs for a more comprehensive evaluation. Secondly, the assumption of 100% medication adherence does not align with real-world scenarios. Future models reflecting varied adherence rates would provide more realistic insights into treatment outcomes and associated costs. Thirdly, the study presumed uniform natural recovery rates after smear microscopy,

which may oversimplify the diversity of patient outcomes. Future research should explore a range of recovery rates to capture real-world variability. Lastly, there was a lack of data on utility parameters before and at the time of diagnosis. Incorporating these parameters in future studies would yield a more understanding of TB's impact on quality of life. Despite the limitations outlined, our study strongly recommends the inclusion of Xpert MTB/RIF in Thailand's health benefit scheme as a replacement for conventional methods such as SSM with culture for TB diagnosis. The cost-effectiveness results affirm this recommendation, particularly highlighting Xpert MTB/RIF Initial's significant potential to reduce TB incidence and mortality rates in line with both national and global TB control strategies. Implementing this policy underscores the necessity for investment in training and capacity-building programs to ensure effective use of Xpert MTB/RIF machines, thereby maximizing their impact on TB control in Thailand.

## Conclusions

The findings of this study suggested that molecular testing methods, whether used as initial diagnostics or as add-on tests, are more cost-effective compared to conventional methods in general Thai populations. Among the algorithms tested, Xpert MTB/RIF Initial emerged as the most cost-effective option. These results provide compelling evidence for policymakers, highlighting the importance of integrating Xpert MTB/RIF Initial into Thailand's health benefit package. This integration is a crucial step toward achieving the aims of the Thailand's national TB strategy and the global End TB Strategy, as it has the potential to significantly reduce TB incidence and mortality rates. The adoption of Xpert MTB/RIF Initial aligns with efforts to enhance TB control and management, particularly in Thailand and other high TB burden countries. Therefore, the findings from our study are highly relevant for policy decision-making in TB management, offering a robust foundation for improving public health outcomes in the context of TB.

## Supporting information

**S1 Table. Model parameters.**
(PDF)

## Acknowledgments

The authors wish to thank Dr. Phalin Kamolwat, Director of the Division of Tuberculosis at the Department of Disease Control, Ministry of Public Health, Thailand and her staff for their great support, provision of epidemiological TB data, and guidance on the data. We also express our appreciation to the team at University of Oxford for their oversight and support in supervising the model.

## Author contributions

**Conceptualization:** Naiyana Praditsitthikorn, Aronrag Meeyai, Jiraphun Jittikoon, Kornkanok Bunwong, Sitaporn Youngkong, Montarat Thavorncharoensap, Surakameth Mahasirimongkol, Wanvisa Udomsinprasert, Usa Chaikledkaew.

**Data curation:** Natthakan Chitpim, Naiyana Praditsitthikorn, Usa Chaikledkaew.

**Formal analysis:** Natthakan Chitpim, Lisa J. White, Aronrag Meeyai, Jiraphun Jittikoon, Kornkanok Bunwong, Sitaporn Youngkong, Montarat Thavorncharoensap, Surakameth Mahasirimongkol, Wanvisa Udomsinprasert, Usa Chaikledkaew.

**Funding acquisition:** Wanvisa Udomsinprasert, Usa Chaikledkaew.

**Investigation:** Natthakan Chitpim, Usa Chaikledkaew.

**Methodology:** Natthakan Chitpim, Naiyana Praditsitthikorn, Lisa J. White, Aronrag Meeyai, Jiraphun Jittikoon, Kornkanok Bunwong, Sitaporn Youngkong, Montarat Thavorncharoensap, Surakameth Mahasirimongkol, Wanvisa Udomsinprasert, Usa Chaikledkaew.

**Project administration:** Usa Chaikledkaew.

**Supervision:** Naiyana Praditsitthikorn, Lisa J. White, Jiraphun Jittikoon, Kornkanok Bunwong, Sitaporn Youngkong, Montarat Thavorncharoensap, Surakameth Mahasirimongkol, Wanvisa Udomsinprasert, Usa Chaikledkaew.

**Validation:** Naiyana Praditsitthikorn, Lisa J. White, Aronrag Meeyai, Jiraphun Jittikoon, Kornkanok Bunwong, Sitaporn Youngkong, Montarat Thavorncharoensap, Surakameth Mahasirimongkol, Wanvisa Udomsinprasert, Usa Chaikledkaew.

**Visualization:** Usa Chaikledkaew.

**Writing – original draft:** Natthakan Chitpim, Naiyana Praditsitthikorn, Lisa J. White, Aronrag Meeyai, Jiraphun Jittikoon, Kornkanok Bunwong, Sitaporn Youngkong, Montarat Thavorncharoensap, Surakameth Mahasirimongkol, Wanvisa Udomsinprasert, Usa Chaikledkaew.

**Writing – review & editing:** Natthakan Chitpim, Naiyana Praditsitthikorn, Lisa J. White, Aronrag Meeyai, Jiraphun Jittikoon, Kornkanok Bunwong, Sitaporn Youngkong, Montarat Thavorncharoensap, Surakameth Mahasirimongkol, Wanvisa Udomsinprasert, Usa Chaikledkaew.

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
