## [Decision Letter · Decision Letter 0]

26 Nov 2024

PONE-D-24-46487Economic evaluation of diagnostic tests for Thai patients with tuberculosis: a dynamic transmission model approachPLOS ONE

Dear Dr. Chaikledkaew,

Thank you for submitting your manuscript to PLOS ONE. After careful consideration, we feel that it has merit but does not fully meet PLOS ONE’s publication criteria as it currently stands. Therefore, we invite you to submit a revised version of the manuscript that addresses the points raised during the review process.

Please submit your revised manuscript by  Jan 10 2025 11:59PM. If you will need significantly more time to complete your revisions, please reply to this message or contact the journal office at plosone@plos.org . Please include the following items when submitting your revised manuscript:

We look forward to receiving your revised manuscript.

Kind regards,

Frederick Quinn

Academic Editor

PLOS ONE

Journal Requirements:

“We would like to acknowledge the Thailand Science Research and Innovation under the Ministry of Higher Education, Science, Research and Innovation for providing the Royal Golden Jubilee Ph.D. scholarship for NC and UC (grant no. PHD/0104/2559). This research project has been funded by the Health System Research Institute (HSRI), Thailand as well as Mahidol University (Fundamental Fund: fiscal year 2024) by National Science Research and Innovation Fund (NSRF). The findings, interpretations and conclusions expressed in this article do not necessarily reflect the views of the aforementioned funding agencies.”

Reviewers' comments:

Reviewer's Responses to Questions

**Comments to the Author**

1. Is the manuscript technically sound, and do the data support the conclusions?

Reviewer #1: Yes

Reviewer #2: Yes

2. Has the statistical analysis been performed appropriately and rigorously? 

Reviewer #1: Yes

Reviewer #2: Yes

3. Have the authors made all data underlying the findings in their manuscript fully available?

Reviewer #1: Yes

Reviewer #2: Yes

4. Is the manuscript presented in an intelligible fashion and written in standard English?

Reviewer #1: Yes

Reviewer #2: Yes

5. Review Comments to the Author

Reviewer #1: The manuscript by Chitpim et al. provides a well-structured analysis of the cost-utility of TB diagnostics in Thailand and has used appropriate methodology. Overall, the results support the main conclusion that molecular tests, such as Xpert, are more cost-effective than conventional methods. The findings offer valuable evidence for policymakers, particularly in advocating for the integration of Xpert MTB/RIF into national health programs.

Additional Comments:

• In line 69, reference number 8 appears to be cited incorrectly.

• Line 132 has a typo: "Tjpse" should be "those."

• Line 369 has a typo: “which have used static models” should be “which has used.”

• While the limitations are well-stated, consider suggesting potential solutions or directions for future research to address these limitations.

Reviewer #2: Grammatical and Typographical Errors

- in Line 132: “Tjpse” should be corrected to “Those.”

- Word Choice in Line 297: “Incremental Costs” might be more consistently phrased as "Incremental Cost" in Table 1, as it’s commonly singular in economic analyses.

-Sentence fragment in Line 296: The phrase “Dominant indicates lower costs and higher QALYs” could be more readable if rephrased to clarify that dominance in this context refers to both lower costs and improved effectiveness or increased QALYs.

6. PLOS authors have the option to publish the peer review history of their article (what does this mean? ). If published, this will include your full peer review and any attached files.

**Do you want your identity to be public for this peer review?** For information about this choice, including consent withdrawal, please see our Privacy Policy .

Reviewer #1: No

Reviewer #2: No

---

## [Author Response · Author response to Decision Letter 0]

29 Nov 2024

November 29, 2024

Dear Editors,

On behalf of my co-authors, we would like to resubmit the revised manuscript of our study entitled “Economic evaluation of diagnostic tests for Thai patients with tuberculosis: a dynamic transmission model approach” (Submission ID PONE-D-24-46487) for your kind consideration to be published on PLOS ONE.

We would like to thank all editors and reviewers for their helpful comments and suggestions. We feel that the revised paper is much further improved as a consequence of their inputs. The next page is a point-by point form explaining how we have responded to each comment raised by the editors and reviewers.

We would like to provide financial statement as follows: “We would like to acknowledge the Thailand Science Research and Innovation under the Ministry of Higher Education, Science, Research and Innovation for providing the Royal Golden Jubilee Ph.D. scholarship for NC and UC (grant no. PHD/0104/2559). This research project has been funded by the Health System Research Institute (HSRI), Thailand as well as Mahidol University (Fundamental Fund: fiscal year 2024) by National Science Research and Innovation Fund (NSRF). The funders had no role in study design, data collection and analysis, decision to publish, or preparation of the manuscript."

All authors declare no competing financial interest. We confirm that the present manuscript is original, not previously published, and not submitted for publication or consideration elsewhere. We also anticipate that you will agree with us on the suitability of this manuscript for publication in PLOS ONE.

Should you have any question, please kindly contact me at usa.chi@mahidol.ac.th. Thank you very much for your kind consideration on this manuscript.

Sincerely yours,

Assoc. Prof. Usa Chaikledkaew, Ph.D.

Corresponding author

Social Administrative Pharmacy Division, Department of Pharmacy and Mahidol University Health Technology Assessment (MUHTA) Graduate Program, Mahidol University

447 Sri-Ayuthaya Road, Rajathevi, Bangkok 10400, Thailand

Tel: 662-644-8679 ext 5317; Fax: 662-644-8694; Email: usa.chi@mahidol.ac.th

Respponses to Editors and Reviewers:

Journal Requirements:

Response: Thank you for your suggestion. We ensure that our manuscript meets PLOS ONE’s style requirements, including those for file naming.

Response: Thank you for your suggestion. All data have been reported in our manuscript and supporting information.

“We would like to acknowledge the Thailand Science Research and Innovation under the Ministry of Higher Education, Science, Research and Innovation for providing the Royal Golden Jubilee Ph.D. scholarship for NC and UC (grant no. PHD/0104/2559). This research project has been funded by the Health System Research Institute (HSRI), Thailand as well as Mahidol University (Fundamental Fund: fiscal year 2024) by National Science Research and Innovation Fund (NSRF). The findings, interpretations and conclusions expressed in this article do not necessarily reflect the views of the aforementioned funding agencies.”

Response: Thank you for your kind suggestion. We have included this amended Role of Funder statement in our cover letter.

Response: Thank you for your helpful suggestions. The ethics statement has been moved to the Methods section (Materials and methods, line 110-117, page 5-6).

Response: Thank you for your helpful suggestions. We have reviewed our reference list and ensured that it is complete and correct. We have already removed the incorrect reference and added the correct reference.

Reviewers' comments

Reviewer #1

The manuscript by Chitpim et al. provides a well-structured analysis of the cost-utility of TB diagnostics in Thailand and has used appropriate methodology. Overall, the results support the main conclusion that molecular tests, such as Xpert, are more cost-effective than conventional methods. The findings offer valuable evidence for policymakers, particularly in advocating for the integration of Xpert MTB/RIF into national health programs.

Response: We would like to thank a reviewer for your valuable comments and suggestions. We feel that the revised paper is much further improved as a consequence of your inputs.

Additional Comments:

• In line 69, reference number 8 appears to be cited incorrectly.

Response: Thank you very much for pointing this out. We apologize for a citing error. We have already deleted reference number 8 and cited reference number 1 as follows (Line 68, page 3).

1. National Tuberculosis Program guideline, Thailand: Bureau of Tuberculosis, Ministry of Public Health; 2018.

• Line 132 has a typo: "Tjpse" should be "those."

Response: Thank you very much for your very helpful suggestion. We apologize for a typing error. We have corrected as suggested (Line 139, page 7).

• Line 369 has a typo: “which have used static models” should be “which has used.”

Response: Thank you very much for your very helpful suggestion. We apologize for a typing error. We have corrected as suggested (Line 374, page 20).

• While the limitations are well-stated, consider suggesting potential solutions or directions for future research to address these limitations.

Response: Thank you very much for your very helpful suggestion. We have revised the paragraph addressing the limitations and directions for future research as follows (Discussion, line 376-386, page 20).

“Firstly, it did not include the costs associated with TB re-diagnosis across all health states, potentially underestimating the total economic burden, especially in cases of treatment failure or relapse. Future studies should incorporate these costs for a more comprehensive evaluation. Secondly, the assumption of 100% medication adherence does not align with real-world scenarios. Future models reflecting varied adherence rates would provide more realistic insights into treatment outcomes and associated costs. Thirdly, the study presumed uniform natural recovery rates after smear microscopy, which may oversimplify the diversity of patient outcomes. Future research should explore a range of recovery rates to capture real-world variability. Lastly, there was a lack of data on utility parameters before and at the time of diagnosis. Incorporating these parameters in future studies would yield a more understanding of TB’s impact on quality of life.”

Reviewer #2

Grammatical and Typographical Errors

Response: We would like to thank a reviewer for your valuable comments and suggestions. We feel that the revised paper is much further improved as a consequence of your inputs.

- in Line 132: “Tjpse” should be corrected to “Those.”

Response: Thank you very much for your very helpful suggestion. We apologize for a typing error. We have corrected as suggested (Line 139, page 7).

- Word Choice in Line 297: “Incremental Costs” might be more consistently phrased as "Incremental Cost" in Table 1, as it’s commonly singular in economic analyses.

Response: Thank you very much for your great suggestion. We have changed from “Incremental Costs” to “Incremental Cost” (Table 1, line 300, page 16).

-Sentence fragment in Line 296: The phrase “Dominant indicates lower costs and higher QALYs” could be more readable if rephrased to clarify that dominance in this context refers to both lower costs and improved effectiveness or increased QALYs.

Response: Thank you very much for your great suggestion. We have rephrased to “Dominance in this context refers to both lower costs and improved effectiveness or increased QALYs.” (Table 1, line 301, page 16).

Response: Thank you for your helpful suggestions. We have already revised figure files and uploaded figures files PACE as suggested.

---

## [Editor Report · Decision Letter 1]

2 Dec 2024

Economic evaluation of diagnostic tests for Thai patients with tuberculosis: a dynamic transmission model approach

PONE-D-24-46487R1

Dear Dr.Chaikledkaew,

We’re pleased to inform you that your manuscript has been judged scientifically suitable for publication and will be formally accepted for publication once it meets all outstanding technical requirements.

Kind regards,

Frederick Quinn

Academic Editor

PLOS ONE
---

## [Editor Report · Acceptance letter]

PONE-D-24-46487R1

PLOS ONE

Dear Dr. Chaikledkaew,

I'm pleased to inform you that your manuscript has been deemed suitable for publication in PLOS ONE. Congratulations! Your manuscript is now being handed over to our production team.

Kind regards,

on behalf of

Dr. Frederick Quinn

Academic Editor

PLOS ONE